# Contribution of Mining Industry in Chosen EU Countries to the Sustainability Issues

**Henrieta Pavolová, Katarína Čulková ***[ID]**, Zuzana Šimková** [ID]**, Andrea Seňová** [ID] **and Dušan Kudelas** [ID]

Institute of Earth's Resources, Faculty of Mining, Ecology, Process Control and Geotechnologies,
Technical University of Kosice, Letná 9, 040 01 Kosice, Slovakia; henrieta.pavolova@tuke.sk (H.P.);
zuzana.simkova@tuke.sk (Z.Š.); andrea.senova@tuke.sk (A.S.); dusan.kudelas@tuke.sk (D.K.)
* Correspondence: katarina.culkova@tuke.sk

**Abstract:** In recent years, the mining industry has achieved an important position in the national economy due to its increasing productivity. However, since 2000, there have been signs of a slowdown, resulting from the national and local conditions of the mining industry. It is for these reasons that we have concluded that this type of industry must be assessed not only from the economic but also from the national and regional sectors, because the performance of the mining industry is directly affected by the value of mineral deposits and the structure of other industries. The present paper aims to analyze the development of the mining industry in Slovakia, in comparison with similar development in chosen European Union countries. Slovakia has been considered as a country with mineral resources and mineral-based products representing an important part of Slovakia's foreign trade, with the significant imported mineral resources including mainly mineral fuels and ore raw materials. The development of the mining industry is assessed from the economics through the growth rate of gross domestic product (GDP) and through the national aspect through the rate of growth of the national economy. The aspects are evaluated by the multi-criteria method Technique for Order of Preference by Similarity to Ideal Solution (TOPSIS), with which we evaluated the country with the best mining industry development. The results of detailed quantitative analyses of the selected indicators for mining industry development for individual European Union countries show a fluctuating trend during the observed period, which is characterized by development disparities. Such results can be used to determine raw material policies in the relevant countries.

**Keywords:** mining industry; energy production; sustainable development; economic indicators; Slovakia



## 1. Introduction

The mining industry, and mining itself, represents a comprehensive set of works needed for the exploration, mining, and also partly the processing of useful minerals and natural resources. The development and performance of industry is conditioned by made decisions by individual economic entities. The industry is an economic sector with different meanings for today's anthropogenic society. On the one hand, it provides the dispersal of local resources and supports short- and long-term prosperity as well as various economic systems; on the other hand, it generates socio-economic disparities and disturbs the environment. The mining industry generally forms a basic platform for the sustainable development of countries, as the extraction of mineral deposits directly determines the development of other national economic sectors. It is for these reasons that we have concluded that this type of industry must be assessed not only from economic but also from national and regional perspectives, as the performance of the mining industry is directly affected by the value of mineral deposits and the structure of other industries.

The mining industry has achieved an important position in national economies [1]. Since 2000, there has been a registered decline of the mining industry due to rising costs [2]. A number of authors have studied the position of the mining industry from different

views. For example, [3] provided a study within the United Nations University (UNU) World Institute for Development Economics Research (WIDER) initiative Extractives for Development. The study shows that mineral resources create a dependency on economic and social development.

Mining industry development influences sustainable development, including society, the economy, and the environment. Another study [4] investigated the contribution of the mining industry to European sustainable development goals. In this area, [5] also examined the interaction of sustainability principles with activities of the mining life cycle. Sustainable access to raw materials has been a growing concern for EU policy since 2008 [6]. Despite efforts to improve sustainable practices of the companies, mining presents a big challenge. To overcome this challenge, mining legislation must be harmonized with other sectors [7]. The social aspect of mining is supported by the Social License to Operate (SLO). The concept of the SLO in relation to mining is still early in its development and acceptance in Europe, possibly due to the very different worldview that exists there [8].

In the EU, major benefits and constraints of mineral extraction have been identified for selected critical raw materials [9]. These benefits are related to securing long-term supplies for the national and strategic important industry sectors. The European Union aims to reduce the import dependency of its industries regarding critical raw materials [10], according to the list of CRM published by the European Commission. There is research demonstrating that sustainability initiatives play an important role in mining companies' operations [11]. Outside the EU, the status of the mineral industry in Sri Lanka has been examined, wherein the establishment of sustainable regulations and policies would enhance the mineral industry [12]. The Indonesian mining sector continues with implications of the mining law requiring certain in-country processing and beneficiation [13].

The above literature confirms that there is a space to solve mining industry development from the view of the concrete state conditions. The present paper aims to evaluate the development of the mining industry in Slovakia, in comparing with similar, chosen states in the EU.

## 2. Present State of Problem Solving

The huge volume of minerals in Slovakia are composed of non-metallic raw materials for construction and energy, which substantially cover their domestic production. Mineral resources and mineral-based products represent an important part of Slovakia's foreign trade. Significant imported mineral resources of Slovakia include mainly mineral fuels (oil, natural gas, hard coal) and ore raw materials (iron ore, raw materials for metallurgy of aluminum, steel and ferroalloys), while exported raw materials produced on a mineral basis include mainly iron and steel, aluminum, ferroalloys, magnesite, cement, bentonite, and dolomite. In 2017, 941 mineral deposits were registered in Slovakia, of which up to 60.5%, i.e., 569 deposits, were exclusive mineral deposits and 39.5%, i.e., 372 deposits, were of non-reserved minerals. In 2017, non-metallic raw materials showed the highest number of exclusive deposits, up to 79.3%, i.e., 451 of the total number of 569 exclusive deposits and 47.9% of the total 941 mineral deposits. The lowest abundance in exclusive deposits was in coal, whose deposits accounted for 3.7% in 2017, i.e., 21 of the total number of exclusive deposits and 2.2% of the registered 941 mineral deposits. A total of 68.2%, i.e., 388 exclusive deposits, consisted of deposits with a designated mining area in a designated protected deposit area; 29.9%, i.e., 170, consisted of deposits with protection without a designated mining area; and only 1.9%, i.e., 11, were other exclusive deposits. In the case of non-reserved mineral deposits, the highest abundance in the analyzed year was recorded by building stone deposits (151 deposits). This accounted for up to 40.6% of the total number of non-reserved mineral deposits and 16.0% of all registered mineral deposits, with the lowest number, only 1.1%, i.e., 4 of the registered deposits of non-reserved minerals, showing limestone deposits, which accounted for only 0.4% of all registered deposits of minerals [14].

From the available data about the exploitation of mineral deposits in Slovakia from the Main Mining Authority [15] of the Slovak Republic and the Slovak Environment Agency, we could state that in 2017, 2955.42 kt of minerals were extracted from the underground and 39,407.00 kt were extracted on the surface, 39 kt of minerals. In the analyzed period (2000–2017), we observed a fluctuating trend in the development of raw mineral exploitation. This presented an average extraction of 31,942.24 kt per year and their highest use in 2015, when about 42,857.04 kt of minerals were extracted, and the lowest in 2003, when only 21,910.32 kt were extracted, which is a decrease of about 58.9% compared to 2015. In general, it can also be stated that the extraction of minerals in 2017 compared to 2000 increased by 68.2% of the original extraction in the observed time horizon, i.e., by approximately 17,184.31 kt, while surface mining of mineral raw materials formed the majority share. In the monitored period, an average of 27,548.16 kt per year was mined on the surface in the monitored period. The highest volume of raw materials was mined on the surface in 2015—approximately 39,908.58 kt—and the least in 2003, when only about 15,967.00 kt were mined, 90 kt of minerals, which represents a decrease of 60.0% compared to 2015. From a global point of view, however, it is possible to state an increase in surface mining of minerals in 2017 by 113.4%, i.e., by about 20,943.49 kt compared to 2000. However, we observed the opposite development trends in underground mining, because in 2017 we observed a decrease in mining by 56.0% compared to 2000, i.e., by 3759.18 kt. In the analyzed period, an average of 4394.08 kt per year of minerals was mined in the analyzed period, with the highest extraction at the level of 6714.60 kt in 2000 and, conversely, the lowest at the level of 2761.35 kt in 2016, which meant a decrease in mining compared to 2000 by 58.9%. In comparing with other V4 countries, Figure 1 shows the situation in the mining industry as follows.

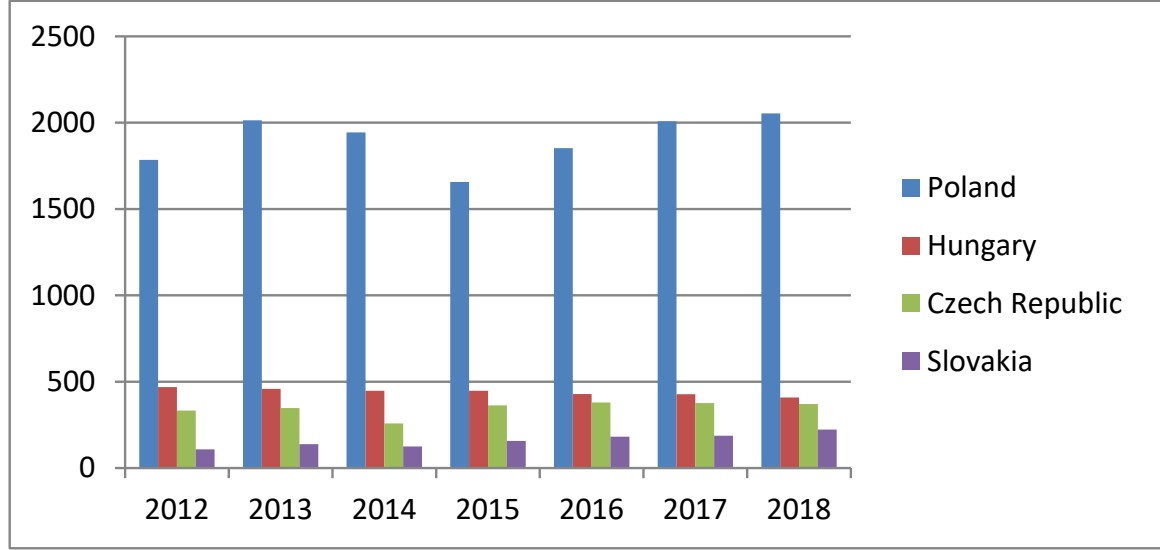

**Figure 1.** Development of mining company numbers in V4.

From Figure 1, it is clear that the number of companies depends on the volume of deposits. The majority of mining companies are in Poland, considerably changing last year. The least number of companies are in Slovakia, with annual growth.

## 3. Materials and Methods

As mentioned above, the mining industry has a special position in the industrial sectors, as its activities provide input materials for other industries. It is for this reason that the very importance of the mining industry is assessed from the point of view of the following basic aspects [16]:

- economic aspects, which means from the view of the mining industry rate at the GDP;

- national aspects, rate at the whole national economy.

The mining industry cannot be characterized only in terms of economic inputs and outputs, because the mining of raw material deposits is specific for significant investment inputs and the production of outputs with relatively low value added, which results mainly from the fact that the mining industry serves as a support type to other industries. It is possible to increase the economic balance of the mining industry only by fundamentally restructuring the mining processes, which, however, is in significant contrast with mining and technical conditions and occupational safety. The share of the mining industry in GDP is very low worldwide, while in EU countries, it is determined by the value of mining and the structure of industries itself [17]. In general, countries can be divided into three basic groups according to the structure of industries and the mining industry, as shown in Table 1.

**Table 1.** Structure of countries according to industry structure.

| Group | Description of the Group |
|---|---|
| Countries with developed mining and less developed other industries | Rich stocks of raw materials with high rate of mining (Iraq, Iran, Venezuela, Botswana, Brazil, Middle and South America, Africa, Arabian Peninsula) |
| Countries with developed mining and other industries | Countries where the mining industry participates at around 5% of the GDP |
| Countries with less developed mining and developed other industries | Rate of the raw material mining is under 1% (small countries without a mining industry, such as Switzerland, where other industries provide high value). |

The economic aspect of mining industry development can be assessed according to the macroeconomic indicators of this industry, which directly affect its performance. Among the basic macroeconomic indicators of the mining industry, development can be included [18]:

- wages, which can generally be characterized as a financial expression of the value of work, resp. the price for work, which is formed in the labor market over time as a result of the effect of supply and demand, which is included in remuneration and payments for work performed;
- employment, which reflects the value of the employed population in a particular country, is one of the key indicators of economic performance;
- labor productivity, which can generally be considered as an economic measure of the use of labor potential expressing the competitiveness and economic performance of industry and as one of the most important factors determining the living standards of the population and the rate of GDP growth;
- import can be described as the total volume of products (goods and services) imported into the territory of the country from abroad, which in practice includes complex values, including licenses, copyright, etc.;
- export can be described as the total volume of products (goods and services) purchased by foreign entities in the financial environment, which in practice includes complex values, including licenses, copyright, etc.;
- turnover can be characterized as value, which means the financial equivalent of the amount of sold (and leased) goods and services (for a certain period). This means the value equivalent of sales, which is in a sense in § 4 par. 7 of the VAT Act defined as revenues without tax on supplied goods and services with the place of delivery in the country.

For the needs of evaluating the development of the mining industry, we used clearly defined indicators, so-called multi-criterial method TOPSIS, with which we evaluated the

country with the best mining industry development. The TOPSIS method is based on the selection of the variant that is closest to the ideal variant and at the same time furthest from the basal variant. In the first step, we created, according to criterial matrix $Y$, normalized criterial matrix $R = (r_{ij})$, resulting from the Equation (1):

$$R_{ij} = \frac{y_{ij}}{\sqrt{\sum_{i=1}^{m} y_{ij}^2}} \tag{1}$$

In the second step, we transformed matrix $R$ to matrix $Z$, by which $j = 1, \dots , n$ is quantified according to Equation (2):

$$z_{ij} = w_j r_{ij} \tag{2}$$

where $w_j$ presents normalized weight for j criteria. Through elements from matrix $Z$, we created the "ideal variant" ($h_1, \dots , h_n$) and "basal variant" ($d_1, \dots , d_n$), in which for $j = 1, \dots ., n$ is quantified following Equations (3) and (4):

$$h_j = \max_{i=1,\dots m} z_{ij} \tag{3}$$

$$d_j = \min_{i=1,\dots m} z_{ij} \tag{4}$$

For any $i = 1, \dots , m$ in the logical consequence; we quantified distances $d^+$, $d^-$ $i$—variant from ideal and basal variant according to Equations (5) and (6):

$$d_i^+ = \sqrt{\sum_{i=1}^{n} \left( z_{ij} - h_j \right)^2} \tag{5}$$

$$d_i^- = \sqrt{\sum_{i=1}^{n} \left( z_{ij} - d_j \right)^2} \tag{6}$$

Furthermore, we determined a relative index of the variant distance from the basal variant according to Equation (7):

$$c_i = \frac{d_i^-}{d_i^+ + d_i^-} \tag{7}$$

In view of the above, we used the TOPSIS variant to rank the variants according to the values of these relative indicators, while the most suitable variant was the one that showed the maximum value, as we solved this method by maximizing.

## 4. Results

Based on detailed quantitative analysis of the development of selected indicators of the mining industry in the individual EU countries monitored, we found that these showed a fluctuating trend during the observed period characterized by development disparities and came to the following partial conclusions:

- We recorded the most significant development disparities in the compared EU countries, especially in employment, employment rates, wages, turnover, imports, etc.
- The highest employment was recorded in Poland with an average employment of 161,747.9 per employee and the lowest in the Slovak Republic with an average employment of 7047.9 per employee, while the Czech Republic reported an average annual employment in the observed period at 30,937.4 employees per year, which was also related to the development of the employment rate with the most significant disparities in the conditions of Slovakia and the smallest in the Czech Republic [19] (see Figure 2).

- Poland spent the most money on the wages of employees in the mining industry, wherein the average total annual wages of employees amounted to EUR 2,999,420 per year. The lowest wages were in Slovakia, in which the average total annual wages of employees amounted to EUR 90,860,000 per year, while in the Czech Republic, it was EUR 496,940,000 per year, which was logically related to the development of employment and thus the rate of employment growth in the analyzed EU countries.
- The development of labor productivity of the mining industry showed milder development disparities between the analyzed countries, with the highest productivity in the conditions of Poland, which showed an average annual value of this monitored indicator at the level of EUR 49,000,000 per year and the lowest in the Czech Republic with an average annual value of EUR 44,300,000 per year, while in Slovakia, it was an average value of EUR 42,900,000 per year [20],
- Poland had the highest turnover again with an average annual value of EUR 311,160.66 thousand per year, the lowest was Slovakia, with an average annual value of EUR 311,160.66 thousand per year, Poland had EUR 4141,75 thousand per year, and the Czech Republic reported an average annual turnover of EUR 28,895.08 thousand per year [21].
- The highest dependence on imports of minerals was shown by Poland, where the average value of imports was EUR 311,160.66 thousand per year, and on the contrary, the lowest dependence was shown in Slovakia, where the average value of imports was EUR 4141.75 thousand per year, while the Czech Republic showed an average import in the analyzed period at the level of EUR 28,895.08 thousand per year [22].
- Exports of minerals showed significantly milder development disparities compared to imports, with the Czech Republic showing the highest exports in 2013–2015 with an average annual value of EUR 377,705.20 thousand per year; in the years 2016–2018, Poland had an average annual value of EUR 357,729.53 thousand per year, while in the conditions of Slovakia, the export of minerals showed an average of EUR 55,491.03 thousand per year [23].
- Selected indicators of the mining industry from the national economic point of view dominated in Poland. This showed the highest share of energy from solid fossil fuels, their share in total available energy, as well as energy from natural gas and renewable energy sources. The lowest values in these indicators were in Slovakia, while the most significant development disparities in the individual analyzed EU countries showed energy obtained from solid fossil fuels and, conversely, the smallest share of solid fossil fuels in the total available energy.

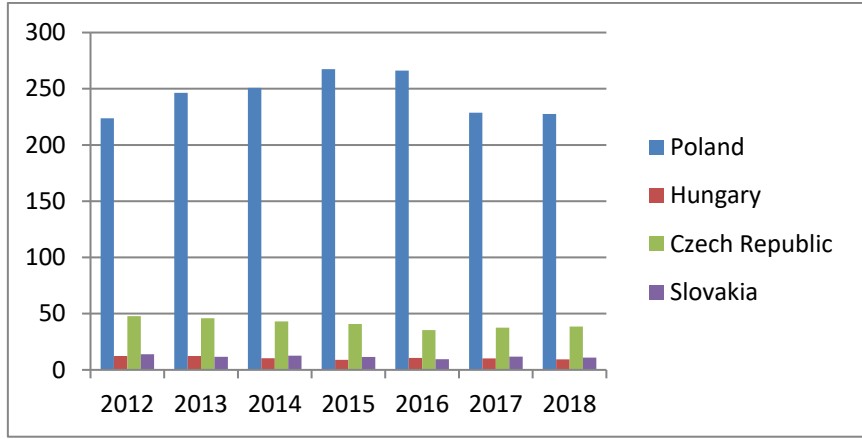

**Figure 2.** Number of employees in mining sector (in thousands of persons).

In view of the above facts, we proceeded to assess the mining industry as an industry of national economies in the individual countries analyzed in terms of a clearly described TOPSIS methodology based on the available values of the last year of the analyzed time.

In terms of the principles and basics of the TOPSIS method, we defined the input criteria according to the analyzed selected indicators from an economic point of view, among which we included six criteria, which were available for analyzed countries, as shown in Table 2.

**Table 2.** Identification of economic criteria.

| Sign | Criteria |
|---|---|
| K1 | Wage |
| K2 | Number of employees |
| K3 | Growth of the employment |
| K4 | Labor productivity |
| K5 | Import |
| K6 | Export |

In accordance with the principles and basics of the TOPSIS method, we constructed an input (Table 3) and auxiliary (Table 4) table of values of the above-mentioned criteria of a mining company in the analyzed EU countries.

**Table 3.** Input values of the criteria from the economic view.

| | K1 | K2 | K3 | K4 | K5 | K6 |
|---|---|---|---|---|---|---|
| Czech Republic | 412.8 | 23.75 | −5.1 | 46.1 | 38.86504 | 321.7052 |
| Poland | 3013.9 | 142.954 | 5.4 | 56 | 323.0311 | 541.8813 |
| Slovakia | 102.3 | 6657 | 6.5 | 46.7 | 5055 | 102.8005 |

**Table 4.** Auxiliary values of the economic criteria.

| | K1 | K2 | K3 | K4 | K5 | K6 |
|---|---|---|---|---|---|---|
| Czech Republic | 170,403.8 | 564.0625 | 26.01 | 2125.21 | 1510.491 | 103,494.2 |
| Poland | 9,083,593 | 20,435.85 | 29.16 | 3136 | 104,349.1 | 293,635.3 |
| Slovakia | 10,465.29 | 44.31565 | 42.25 | 2180.89 | 25.55303 | 10,567.94 |
| Square root (Sum) | 3043.758 | 145.0663 | 9.870157 | 86.26761 | 325.4 | 638.5119 |

Subsequently, we constructed matrix *R* (Table 5) and matrix *Z* (Table 6) for equally evaluated scales at the level of 1/6, as we accepted their mutual interaction in the multi-criteria assessment of the mining industry.

**Table 5.** R-matrix for the economic criteria.

| | K1 | K2 | K3 | K4 | K5 | K6 |
|---|---|---|---|---|---|---|
| Czech Republic | 55.98469 | 3.888309 | 2.635216 | 24.63509 | 4.641952 | 162.0866 |
| Poland | 2984.335 | 140.8725 | 2.95436 | 36.352 | 320.6795 | 459.8744 |
| Slovakia | 3.438279 | 0.305486 | 4.28058 | 25.28052 | 0.078528 | 16.55089 |

**Table 6.** Z-matrix for the economic criteria.

| | K1 | K2 | K3 | K4 | K5 | K6 |
|---|---|---|---|---|---|---|
| Czech Republic | 9.330781 | 0.648051 | 0.439203 | 4.105848 | 0.773659 | 27.01444 |
| Poland | 497.3892 | 23.47875 | 0.492393 | 6.058667 | 53.44659 | 76.64574 |
| Slovakia | 0.573047 | 0.050914 | 0.71343 | 4.21342 | 0.013088 | 2.758481 |
| $h_j$ | 497.3892 | 23.47875 | 0.71343 | 6.058667 | 53.44659 | 76.64574 |
| $d_j$ | 0.573047 | 0.050914 | 0.439203 | 4.105848 | 0.013088 | 2.758481 |

For any criterion, we quantified in distance $d_i{}^+$, $d_i{}^-$ $i$ variants from ideal and basal variants (Table 7) and weight entropy for specific criteria (Table 8).

**Table 7.** Distance quantification.

|  | $d_i+$ | $d_i-$ | $c_i$ | Rank |
|---|---|---|---|---|
| Czech Republic | 493.9269 | 25.80667 | 0.049654 | 2 |
| Poland | 0.221037 | 505.6613 | 0.999563 | 1 |
| Slovakia | 505.6609 | 0.294572 | 0.000582 | 3 |

**Table 8.** Weight entropy for mining industry criteria from the economic view.

|  | K1 | K2 | K3 | K4 | K5 | K6 |
|---|---|---|---|---|---|---|
| Czech Republic | 412.8 | 23.75 | 0.1 | 46.1 | 38.86504 | 321.7052 |
| Poland | 3013.9 | 142.954 | 10.6 | 56.0 | 323.0311 | 541.8813 |
| Slovakia | 102.3 | 6.657 | 11.7 | 46.7 | 5.055 | 102.8005 |
| sum | 3529 | 173.361 | 22.4 | 148.8 | 366.9512 | 966.387 |
| Czech Republic | 0.116974 | 0.136997 | 0.004464286 | 0.309812 | 0.105913 | 0.332895 |
| Poland | 0.854038 | 0.824603 | 0.473214286 | 0.376344 | 0.880311 | 0.560729 |
| Slovakia | 0.028988 | 0.0384 | 0.522321429 | 0.313844 | 0.013776 | 0.106376 |

Subsequently, we were able to explicitly quantify the values of the weights of the above-defined criteria for the development of the mining industry in selected EU countries (Table 9, weights rounded).

**Table 9.** Weight calculation for the economic criteria.

|  | K1 | K2 | K3 | K4 | K5 | K6 |
|---|---|---|---|---|---|---|
| Czech Republic | −0.251 | −0.27232 | −0.024159134 | −0.36303 | −0.23779 | −0.36616 |
| Poland | −0.13475 | −0.15903 | −0.354062221 | −0.36778 | −0.11222 | −0.32439 |
| Slovakia | −0.10264 | −0.12517 | −0.339233204 | −0.3637 | −0.05903 | −0.23836 |
| Sum | −0.4884 | −0.55652 | −0.717454559 | −1.09452 | −0.40904 | −0.92892 |
|  | 0.444558 | 0.506568 | 0.653055283 | 0.996273 | 0.372323 | 0.845537 |
|  | 0.555442 | 0.493432 | 0.346944717 | 0.003727 | 0.627677 | 0.154463 |
| Weight | 0.254593 | 0.22617 | 0.159025927 | 0.001708 | 0.287703 | 0.0708 |

From the final constructed matrix of weights of clearly defined criteria of the mining industry from an economic point of view, we could state that in terms of the evaluated multi-quantitative set of selected criteria, Poland has the most advanced mining industry and Slovakia has the least developed, while the Czech mining industry is between the two countries (Table 10).

**Table 10.** Final Z-matrix for economic criteria weights according to the entropy.

|  | K1 | K2 | K3 | K4 | K5 | K6 |
|---|---|---|---|---|---|---|
| Czech Republic | 14.25332 | 0.14657 | 0.069844623 | 0.007013 | 0.222584 | 1.912622 |
| Poland | 759.7912 | 5.310192 | 0.078303314 | 0.010349 | 15.37673 | 5.426517 |
| Slovakia | 0.875362 | 0.011515 | 0.113453876 | 0.007197 | 0.003765 | 0.1953 |
| $h_j$ | 759.7912 | 5.310192 | 0.113453876 | 0.010349 | 15.37673 | 5.426517 |
| $d_j$ | 0.875362 | 0.011515 | 0.069844623 | 0.007013 | 0.003765 | 0.1953 |

|  | $d_i+$ | $d_i-$ | $C_i$ | Rank |
|---|---|---|---|---|
| Czech Republic | 493.9269 | 25.80667 | 0.049654 | 2 |
| Poland | 0.221037 | 505.6613 | 0.999563 | 1 |
| Slovakia | 505.6609 | 0.294572 | 0.000582 | 3 |

Analogically, we calculated other indicators, including the national economic view in chosen EU countries, which means we defined input criteria according to the analyzed chosen indicators of the mining industry, including six criteria (selected according to the mining activities in the analyzed countries), as mentioned in Table 11.

**Table 11.** Identification of other criteria.

| Sign | Title of the Criteria |
|---|---|
| K1 | Solid fossil fuels |
| K2 | Rate of solid fossil fuels on total available energy volume |
| K3 | Earth gas—energy |
| K4 | Labor productivity |
| K5 | Renewable energy sources |
| K6 | Turnover in mining industry |

According to the principles and basics of the TOPSIS methods, we constructed the input table (Table 12) and table with auxiliary values (Table 13) of the values for specific criteria of the mining industry in the analyzed EU countries.

**Table 12.** Input values of other criteria.

| | K1 | K2 | K3 | K4 | K5 | K6 |
|---|---|---|---|---|---|---|
| Czech Republic | 165,602.7 | 73.59 | 83,413.54 | 57,139.21 | 16.24 | 4938.1 |
| Poland | 509,869.7 | 89.6 | 196,833.5 | 115,189.4 | 12.16 | 12,532.7 |
| Slovakia | 31,749.27 | 62.41 | 47,522.78 | 25,626.51 | 16.89 | 607.7 |

**Table 13.** Auxiliary values of other criteria.

| | K1 | K2 | K3 | K4 | K5 | K6 |
|---|---|---|---|---|---|---|
| Czech Republic | $2.74 \times 10^9$ | 5415.488 | $6.96 \times 10^9$ | $3.26 \times 10^9$ | 263.7376 | 24,384,832 |
| Poland | $2.6 \times 10^{11}$ | 8028.16 | $3.87 \times 10^{10}$ | $1.33 \times 10^{10}$ | 147.8656 | $1.57 \times 10^8$ |
| Slovakia | $1.01 \times 10^9$ | 3895.008 | $2.26 \times 10^9$ | $6.57 \times 10^8$ | 285.2721 | 369,299.3 |
| Square root (Sum) | 537,028.3 | 131.6763 | 218,997 | 131,111.4 | 26.3984 | 13,484.16 |

Consequently, we constructed the R-matrix (Table 14) and Z-matrix (Table 15) for similarly evaluated weights at the level 1/6, since we accepted their mutual interaction during multi-criteria evaluation of the mining industry.

**Table 14.** R-matrix for other criteria.

| | K1 | K2 | K3 | K4 | K5 | K6 |
|---|---|---|---|---|---|---|
| Czech Republic | 51,066.65 | 41.12727 | 31,771.3 | 24,901.64 | 9.990668 | 1808.405 |
| Poland | 484,084.6 | 60.96889 | 176,913.1 | 101,200.9 | 5.60131 | 11,648.37 |
| Slovakia | 1877.026 | 29.58017 | 10,312.54 | 5008.855 | 10.80642 | 27.38763 |

**Table 15.** Z-matrix for other criteria.

| | K1 | K2 | K3 | K4 | K5 | K6 |
|---|---|---|---|---|---|---|
| Czech Republic | 8511.109 | 6.854545 | 5295.217 | 4150.274 | 1.665111 | 301.4008 |
| Poland | 80,680.77 | 10.16148 | 29,485.52 | 1686.,82 | 0.933552 | 1941.395 |
| Slovakia | 312.8377 | 4.930028 | 1718.756 | 834.8092 | 1.80107 | 4.564605 |
| $h_j$ | 80,680.77 | 10.16148 | 29,485.52 | 16,866.82 | 1.80107 | 1941.395 |
| $d_j$ | 312.8377 | 4.930028 | 1718.756 | 834.8092 | 0.933552 | 4.564605 |

For any criteria, we quantified in distance $d_i{}^+$, $d_i{}^-$ $i$ variants from the ideal and basal variant (Table 16) and weight entropy for higher mentioned criteria (Table 17).

**Table 16.** Quantification of distance for other criteria.

|  | $d_i+$ | $d_i-$ | $C_i$ | Rank |
|---|---|---|---|---|
| Czech Republic | 77,188.28 | 9543.749 | 0.110037 | 2 |
| Poland | 0.867518 | 86,549.26 | 0.99999 | 1 |
| Slovakia | 86,549.26 | 0.867518 | 0.00001 | 3 |

**Table 17.** Weight entropy for other criteria of mining industry.

|  | **K1** | **K2** | **K3** | **K4** | **K5** | **K6** |
|---|---|---|---|---|---|---|
| Czech Republic | 165,602.7 | 73.59 | 83,413.54 | 57,139.21 | 16.24 | 4938.1 |
| Poland | 509,869.7 | 89.6 | 196,833.52 | 115,189.4 | 12.16 | 12,532.7 |
| Slovakia | 31,749.27 | 62.41 | 47,522.78 | 25,626.51 | 16.89 | 607.7 |
| sum | 707,221.7 | 225.6 | 327,769.84 | 197,955.1 | 45.29 | 18,078.5 |
| Czech Republic | 0.234159 | 0.326197 | 0.254488149 | 0.288647 | 0.358578 | 0.273148 |
| Poland | 0.720948 | 0.397163 | 0.600523587 | 0.581896 | 0.268492 | 0.693238 |
| Slovakia | 0.044893 | 0.27664 | 0.144988264 | 0.129456 | 0.37293 | 0.033615 |

Consequently, we explicitly quantified values of the weights for specific criteria of the mining industry development in chosen EU countries (Table 18, weights rounded).

**Table 18.** Weight calculation for other criteria.

|  | **K1** | **K2** | **K3** | **K4** | **K5** | **K6** |
|---|---|---|---|---|---|---|
| Czech Republic | −0.33994 | −0.36542 | −0.348267289 | −0.35866 | −0.36776 | −0.35448 |
| Poland | −0.23589 | −0.36674 | −0.306239021 | −0.31508 | −0.35305 | −0.25399 |
| Slovakia | −0.13932 | −0.35549 | −0.279987196 | −0.26466 | −0.36784 | −0.11405 |
| Sum | −0.71515 | −1.08766 | −0.934493505 | −0.9384 | −1.08866 | −0.72251 |
|  | 0.650959 | 0.990031 | 0.850612645 | 0.854165 | 0.990937 | 0.657659 |
|  | 0.349041 | 0.009969 | 0.149387355 | 0.145835 | 0.009063 | 0.342341 |
| Weights | 0.347084 | 0.009913 | 0.148550019 | 0.145018 | 0.009013 | 0.340422 |

From the next matrix with weights, we can clearly define other criteria of the mining industry. Including indicators from the national economic view, we can state that in the sense of the evaluated multi-quantitative set of the other chosen criteria, the most developed mining industry is in Poland and the least developed is in Slovakia, while the Czech mining industry belongs somewhere between the two (see Table 19).

**Table 19.** Final Z-matrix for the other criteria weights according to the entropy.

|  | **K1** | **K2** | **K3** | **K4** | **K5** | **K6** |
|---|---|---|---|---|---|---|
| Czech Republic | 17,724.43 | 0.067951 | 786.6046023 | 601.8642 | 0.015007 | 102.6034 |
| Poland | 168,018.2 | 0.100733 | 4380.074485 | 2445.991 | 0.008414 | 660.8933 |
| Slovakia | 651.4863 | 0.048873 | 255.3213042 | 121.0623 | 0.016232 | 1.553891 |
| $h_j$ | 168,018.2 | 0.100733 | 4380.074485 | 2445.991 | 0.016232 | 660.8933 |
| $d_j$ | 651.4863 | 0.048873 | 255.3213042 | 121.0623 | 0.008414 | 1.553891 |

|  | $d_i+$ | $d_i-$ | $C_i$ | Rank |
|---|---|---|---|---|
| Czech Republic | 150,349 | 17,088.28 | 0.102057751 | 2 |
| Poland | 0.007819 | 167,435 | 0.999999953 | 1 |
| Slovakia | 167.435 | 0.007819 | 0.000000047 | 3 |

According to the results of the TOPSIS method, we can state that the mining industry registered significant differences in the analyzed EU countries, while its development was solidly determined by a single value of the mining and its structure in the national economy; the most effective mining industry is in Poland and the least effective conditions of mining industry are in Slovakia.

## 5. Discussions and Conclusions

According to the quantitative analysis of the chosen indicators of mining industry development in the individual analyzed EU countries, we found that indicators had a fluctuant trend in the development during the analyzed time. The trend is characterized by developing disparities, while the most considerable developing disparities were registered in the compared EU countries, mostly from the view of employment, measure of employment, wages, turnover, and import; the least significant developing disparities were in labor productivity. The highest dependence on raw material imports was registered in Poland, where the average value of the import presented EUR 311,160.66 thousand/year. On the other hand, the least dependence is registered in Slovakia, where the average value of the import presented EUR 414,175,000/year, while the Czech Republic recorded in the analyzed period an average import at the level EUR 2,889,508,000/year. The export of raw materials recorded compared with the import showed considerably lower developing disparities, while the highest export was registered in the Czech Republic between 2013 and 2015 at an average value of EUR 37,770,520,000/year, and in 2016–2018 in Poland with an average annual value at the level EUR 35,772,953,000/year. In Slovakia, the raw material export was EUR 5,549,103,000/year. At the same time, we found that Poland dominated in chosen indicators of its mining industry from the national economic view, where it achieved the highest rate of energy, obtained from solid fossil fuels, as well as rate on total available energy and energy, obtained from earth gas and renewable energy sources. The lowest values of the mentioned indicators were in Slovakia, with the most significant developing disparities in the individual analyzed EU countries recorded for energy obtained from solid fossil fuels. On the other hand, the lowest value was the rate of solid fossil fuels on total available energy. According to the results of the multi-criteria TOPSIS method, we found that in the sense of the evaluated multi-quantitative set of chosen economic criteria, the most developed mining industry is in Poland and the least developed mining industry is in Slovakia. However, some of the identified first places of the chosen indicators of the mining industry development have a more negative than positive impact, for instance, Poland's high dependence on imports. Slovakia has a rich and vast range of raw materials, but only a low volume is mined. If its raw material deposit extraction was more intensive, Slovakia could obtain higher independence on imports, which would consequently increase security and independence. This could lead to the ability to reduce developing disparities, mostly in comparing with the Czech Republic.

Since V4 countries are post-communist countries, their mining industry has similar development and problems. However, it is necessary to study the differences, reflected in the energy policy and security [24]. The results of the contribution are useful for the creation of raw material policy [25,26], but also for local communities, sustainable development of the regions, and the society [27–30].

**Author Contributions:** Conceptualization, H.P. and K.Č.; methodology, H.P. and Z.Š.; software, A.S.; validation, D.K.; formal analysis, H.P. and Z.Š.; resources, K.Č.; data curation, A.S.; writing—original draft preparation, K.Č.; writing—review and editing, K.Č. and A.S.; visualization, H.P.; supervision, D.K.; project administration, K.Č.; funding acquisition, D.K. All authors have read and agreed to the published version of the manuscript.

**Funding:** This research was funded by VEGA 1/0797/20.

**Institutional Review Board Statement:** Not applicable.

**Informed Consent Statement:** Not applicable.

**Acknowledgments:** Contribution presents partial result of grant project VEGA 1/0797/20 and VEGA 1/0264/21.

**Conflicts of Interest:** The authors declare no conflict of interest.

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
