# Peer review of "Contribution of Mining Industry in Chosen EU Countries to the Sustainability Issues"

_sustainability, doi:10.3390/su14074177_

Round 1

Reviewer 1 Report

The authors in the article present currently a very important problem connected with the use of energy production energy raw materials excavated underground. The topic issue in the article is to search mining industry development in Slovakian conditions, in comparing with the similar states in chosen EU countries, mainly Poland and the Chech Republic. The development of the mining industry in Slovakia was assessed from the point of economic aspects through the rate at the GDP and national aspects through the rate at the whole national economy. The obtained results of the quantitative analysis of the selected indicators of the development in the EU mining industry allow showing the variable tendency in the observed period. The received results can be used for the determination of raw material policy in the selected countries.
The article was written clearly, beyond the form of presenting the results.  There were found also a few faults in the edition form, which should be corrected. There are attached below. 

Single letters at the end of the line and single words at the end of the paragraph should be corrected in the text - lines 35, 165, 173, 192 and 379
Line 194 - correct the format of the word "so-called", remove the superscript,
I propose in the text to refer to the numbers of the equations, e.g. line 198 - equation (1)
In tables 7, 9, 10, 16, 18 and 19, some cells are marked with a colour. Is that necessary? - it introduces some inconsistency in the presentation of the results.
Tables 10, 15 and 19 should be presented in full on one page.
Instead of the form "EUR.year-1," on lines 354, 356, 360, 361 and 362 I propose the form "EUR per year" or "EUR / year".
The authors present the obtained results only in tables (19 in total). This form of presentation does not allow for quick comparison and evaluation of the obtained results. I propose to replace the selected results in the tables graphically, in the form of charts, e.g. bar charts or diagrams, if possible. This will make it easier for the reader to read the article.

Author Response

Dear reviewer, thank you for your review. We considered all your recommendations an suggested correction. The lines 35, etc. is corrected, format so-called is corrected, we add number of equation. As for the color of the tables – not necessary, due to the inconsistency we changed it.

Table 10,15,19 is corrected to one page.

We corrected EUR/year in all text. As for your suggestion to make figures illustrations, we added some, however some tables presents matrix, which is already graphical presentation, so not all tables can be expressed in figure.

Thank you once again for your review. Best regards, K.Culkova, corresponding author

Reviewer 2 Report

  1. Results
  2. “In terms of the principles and principles of the TOPSIS method, we defined the input criteria according to the analyzed selected indicators from an economic point of view, among which we included six criteria as we saw in Table 2.” Why only analyze these six criteria, what is the basis for the selection of these six criteria? Please give a specific explanation.

  1. “Table 9. Weights calculation for the economic criteria.” In the Table 9, the sum of weights calculated of K1 to K6 is less 1. Please give specific reasons and corresponding explanations. Whether the TOPSIS method of calculating the weight is correct needs further verification.

  1. “Table 11. Identification of other criteria.”What is the basis for the identification of other criteria? Please give a specific explanation.

  1. “Table 18. Weights calculation for other criteria.”In the Table 18, the sum of weights calculated of K1 to K6 is over 1. Please give specific reasons and corresponding explanations. Whether the TOPSIS method of calculating the weight is correct needs further verification.

References

  1. References format was not unified. And some issue number or volume number or page number of references were missed. In a word, the references exist many mistakes, please modified them. Additionally, references need to be arranged alphabetically, please correct them carefully.

Author Response

Dear reviewer, thank you very much for your review, we appreciate your notes and suggestions for corrections which can increase quality of the presented paper. As for your individual comments, we answer as follows:

  1. Why only analyze these six criteria, what is the basis for the selection of these six criteria? Please give a specific explanation. We resulted from the available data for any analysed country. Due to the different availability we have only six common criteria available.
  2. Table 9 and table 18 – due to the big number behind decimal point we rounded the weights
  3. Table 11 - Identification of other criteria.”What is the basis for the identification of other criteria? Please give a specific explanation. We made it according to the mining activities in the analysed countries
  4. References is modified, made alphabetically and according to the template of the journal

Thank you once again for your review. Best regards, K.Culkova, corresponding author